# A data-driven prospective study of dementia among older adults in the United States

**Jordan Weiss** [1¤]*, **Eli Puterman** [2], **Aric A. Prather** [3], **Erin B. Ware** [4], **David H. Rehkopf** [5]*

**1** Population Studies Center and the Leonard Davis Institute of Health Economics, University of Pennsylvania, Philadelphia, Pennsylvania, United States of America, **2** School of Kinesiology, The University of British Columbia, Vancouver, British Columbia, Canada, **3** Department of Psychiatry, University of California, San Francisco, San Francisco, California, United States of America, **4** Survey Research Center, Institute for Social Research, University of Michigan, Ann Arbor, Michigan, United States of America, **5** School of Medicine, Stanford University, Palo Alto, California, United States of America

¤ Current address: Department of Demography, University of California, Berkeley, Berkeley, California, United States of America

* drehkopf@stanford.edu (DHR); jordanmnw@gmail.com (JW)

## Abstract

### Background

Studies examining risk factors for dementia have typically focused on testing a priori hypotheses within specific risk factor domains, leaving unanswered the question of what risk factors across broad and diverse research fields may be most important to predicting dementia. We examined the relative importance of 65 sociodemographic, early-life, economic, health and behavioral, social, and genetic risk factors across the life course in predicting incident dementia and how these rankings may vary across racial/ethnic (non-Hispanic white and black) and gender (men and women) groups.

### Methods and findings

We conducted a prospective analysis of dementia and its association with 65 risk factors in a sample of 7,908 adults aged 51 years and older from the nationally representative US-based Health and Retirement Study. We used traditional survival analysis methods (Fine and Gray models) and a data-driven approach (random survival forests for competing risks) which allowed us to account for the semi-competing risk of death with up to 14 years of follow-up. Overall, the top five predictors across all groups were lower education, loneliness, lower wealth and income, and lower self-reported health. However, we observed variation in the leading predictors of dementia across racial/ethnic and gender groups such that at most four risk factors were consistently observed in the top ten predictors across the four demographic strata (non-Hispanic white men, non-Hispanic white women, non-Hispanic black men, non-Hispanic black women).

### Conclusions

We identified leading risk factors across racial/ethnic and gender groups that predict incident dementia over a 14-year period among a nationally representative sample of US aged 51 years and older. Our ranked lists may be useful for guiding future observational and

**Data Availability Statement:** This study analyzes publicly available data from Health and Retirement Study. Persons interested in obtaining data files from the Health and Retirement Study should access the Health and Retirement Study's Data

Products Database (https://hrs.isr.umich.edu/data-products). The authors did not receive special access privileges to the data that others would not have.

**Funding:** The author(s) received no specific funding for this work.

**Competing interests:** The authors have declared that no competing interests exist.

**Abbreviations:** ADAMS, Aging, Demographics, and Memory Study; CI, Confidence interval; HR, hazard ratio; HRS, Health and Retirement Study; NIA, National Institute on Aging; PGS, Polygenic score; sdHR, subdistribution hazard ratio; US, United States.

quasi-experimental research that investigates understudied domains of risk and emphasizes life course economic and health conditions as well as disparities therein.

## Introduction

In 2017, the Lancet Commission on Dementia Prevention and Care published a report to consolidate the state of knowledge on preventive and management strategies for cognitive dementia [1]. The Commission reviewed evidence from over 500 scientific peer-reviewed articles, systematic reviews, and meta-analyses and calculated that nearly one third of dementia cases may be preventable. A wide range of factors may contribute to the ability to prevent one third of these cases including educational attainment, social engagement, physical activity, and management of comorbidities.

However, a majority of the studies reviewed in the Commission's report examined risk factors independent of one another to test *a priori* hypotheses about how they may be associated with dementia. Few studies have jointly and comparatively analyzed risk factors for dementia across domains of sociodemographic, early-life, economic, health and behavioral, social, and genetic characteristics while also systematically examining whether these factors differ among racial/ethnic and gender groups. A study by Lourida and colleagues [2] reported that a favorable lifestyle (e.g., not being an active smoker, engaging in regular physical activity, and maintaining a healthy diet) was associated with a lower dementia risk irrespective of genetic risk for dementia in a sample of more than 190,000 participants of European ancestry. Another recent study [3] reported that sociodemographic characteristics (e.g., lower educational attainment, Hispanic origin) and measures of health (e.g., lower rated subjective health, higher levels of body mass index [BMI]) were comparatively better predictors of incident dementia than genetic risk of dementia assessed through polygenic scores. Together, these studies suggest that a healthy lifestyle may help offset the genetic risk of dementia.

Despite these promising findings, there has been limited work integrating risk factors across multiple domains to understand their relative importance for predicting dementia and how these rankings may vary across racial/ethnic and gender groups. An analytic framework that allows for a comprehensive investigation of dementia risk factors may be useful for hypothesis generation and prioritizing group-specific intervention targets to prevent or delay the onset of dementia [4, 5]. Further, this may help shape our understanding of how intervening on specific risk factors may eradicate or exacerbate documented disparities in dementia risk [6, 7].

Prior studies in which investigators used more contemporary statistical approaches to examine dementia have focused primarily on medical risk factors [8] and neuroimaging biomarkers [9] as well as resilience to genetic predispositions to dementia [10]. Although these characteristics are important to studying the onset of dementia, little work [e.g., 3, 11, 12] has combined genetic and life-course environmental risk factors in pursuit of a more comprehensive prediction model. A study by Casanova and colleagues [11] used data from the Health and Retirement Study (HRS) combined with a data-driven approach to predict cognitive impairment using sociodemographic, health, and genetic data. These researchers found that education, age, gender, and history of stroke were among the leading characteristics predicting cognitive impairment. Despite the novelty and innovation of their approach, the authors did not account for the semi-competing risk of death nor did they examine differential rankings of predictors by race/ethnic and gender. Failure to account for the semi-competing risk of

mortality when studying older populations can bias results and overestimate the risk of disease [13]. In addition, documented differences in longevity and dementia incidence among race/ethnicity and gender groups could bias results as they absorb much of the variation in this prior study. Due to data limitations, this prior study also examined a smaller list of predictors, for example, using neighborhood socioeconomic status rather than a range of social and economic factors at the individual level. Sapkota and colleagues [12] used a data-driven approach to test the predictive importance of 19 characteristics from six risk factor domains (novel metabolomics biomarker panels; selected Alzheimer's disease genetic risk polymorphisms; functional health; lifestyle engagement; cognitive performance; and biodemographic factors) for mild cognitive impairment and Alzheimer's disease. The authors reported that characteristics from multiple risk factor domains were important for classifying mild cognitive impairment and Alzheimer's disease; however, their analysis was limited to a sample of fewer than 100 respondents. More recently, Aschwanden and colleagues [3] conducted a similar study using the HRS but did not account for the semi-competing risk of death nor did they investigate variation across racial/ethnic and gender groups.

Understanding the relative importance and predictive power of these factors remains understudied and is critical for planning group-specific treatment strategies for those who may be at greater risk of dementia. We build on this emerging literature by investigating the relative importance of 65 early-life, sociodemographic, early-life, economic, health and behavioral, social, and genetic characteristics across the life course to dementia in the nationally representative and longitudinal US-based Health and Retirement Study (HRS). We estimated hazard ratios (HRs) of each characteristic for incident dementia while accounting for the semi-competing risk of death. We then compared these results to those obtained within a data-driven framework by utilizing random survival forests for competing risks. All models were stratified by race/ethnicity and gender to examine the differential ranking of each predictor across these demographic strata.

## Methods

### Study population

The HRS is a nationally representative and longitudinal study of more than 30,000 community-dwelling US adults aged 51 years and older and their spouses of any age. Since 1992, the HRS has biennially collected economic, social, and health information from respondents who undergo detailed telephone or in-person interviews. Respondents who are unable or unwilling to participate may be surveyed by a proxy respondent, typically a spouse or adult child, who completes the survey on their behalf. The HRS is under current IRB approval at the University of Michigan and the National Institute on Aging (NIA) with support from the NIA (NIA; U01AG009740) and the Social Security Administration [14]. Polygenic score data were available for HRS respondents who provided written informed consent consented and provided salivary DNA from 2006 through 2012; respondents who did not sign the consent form were not asked to complete the collection [15]. All data used in this study are de-identified and publicly available.

We used a base year of 2000 with follow-up through 2014 during which time cognitive information was consistently ascertained for community-dwelling and nursing home residents. We restricted the analytic sample to non-Hispanic men and women aged 51 years and older who were dementia-free at baseline in 2000 who had polygenic score data, a valid sampling weight, and at least one measure of cognitive function over the study period (2000 to 2014). We further excluded respondents who self-reported their race as "Other Race" due to low sample sizes.

## Measures

**Outcome.** Different protocols were used to assess cognitive function among self- and proxy- respondents in the HRS [16]. Among self-respondents, cognitive function was determined through a series of cognitive tests which included immediate and delayed 10-noun free recall tests (range: 0–10 points each), a serial 7s subtraction test (range: 0–5 points), and a backwards counting test (range: 0–2 points). Scores ranged from 0 to 27, with higher scores reflecting better cognitive performance. Among respondents surveyed through a proxy, cognitive scores were based on the proxy's assessment of the respondent's memory (range: 0–4; excellent, very good, good, fair, poor), limitations in five instrumental activities of daily living (range: 0–5; managing money, taking medication, preparing hot meals, using phones and doing groceries), and the interviewer's assessment of the respondent's difficulty completing the interview due to cognitive limitations (range: 0–2; none, some, prevents completion) to produce a score ranging from 0 to 11, with higher scores reflecting a higher degree of impairment. Detailed information about the cognitive assessments are publicly available and provided by the HRS investigators [16].

Cut points for all-cause dementia using these scales in the HRS were validated against the Aging, Demographics, and Memory Study (ADAMS). The ADAMS is a clinical substudy of 856 HRS respondents who underwent extensive in-home neuropsychological and clinical assessments [17]. We used the Langa-Weir approach [18] in our primary analysis to classify respondents with dementia (self-respondent: 0–6 out of 27; proxy: 6–11 out of 11). Dementia status for self- and proxy-respondents was assessed at each survey wave.

In sensitivity analyses, we used three additional classification schemes for dementia which are reported to have greater sensitivity to racial/ethnic and sociodemographic disparities [19, 20]. These alternative schemes, referred to as the Hurd Model, the Expert Model, and the LASSO Model were also validated against the ADAMS as described elsewhere [20]. Our sensitivity analyses were conducted in a subsample of respondents who, in addition to the inclusion criteria for the analytic sample, were 70 years or older at baseline and had available information on all four dementia classification methods (which were estimated among HRS respondents aged 70+ years).

## Risk factors

We conducted a thorough review of the articles cited in the Lancet Commission's report and selected 65 risk factors that were available in the HRS. We classified risk factors into seven domains: sociodemographic (1), early-life (2), economic (3), health (4), behavioral (5), social ties (6), and genetic markers (7). A complete list of risk factors, their definition, and coding is provided in the S1 Appendix in S1 File. All risk factors measured on a continuous scale were standardized to a normal distribution (mean = 0, standard deviation = 1). Binary variables were coded -1 and 1 to improve comparability with the continuous measures standardized to a normal distribution. Risk factors were coded such that higher scores reflected a higher degree of risk. All risk factors were measured in 1998 or 2000.

## Statistical analysis

All statistical analysis was performed in R version 3.6.1 [21]. All analyses used respondent-level sampling weights and, where appropriate, included robust standard errors to account for the clustering of individuals within households in the HRS. In preparing the data file, we excluded risk factors that were missing among 20% or more of respondents (see S1 Table in S1 File). Missing data values for the remaining predictors were imputed using a non-parametric approach implemented with the R package 'missForest' with five iterations each fit with 500

trees [22]. We examined associations between all predictors by creating correlation matrices for all risk factors across racial/ethnic and gender groups. The distribution of all 65 risk factors at baseline was examined by computing the prevalence or mean and standard deviation of each risk factor after imputation.

We used inverse probability weighting to account for selection into the HRS genetic sample. This process upweighted respondents with a lower propensity for providing genetic data, creating a pseudo population which more closely reflects the representativeness of the HRS sample [23, 24]. The respondent-level sampling weights used to generate new base weights for our analysis were calculated and provided by the HRS investigators. Specifically, we used the respondent-level sampling weights that account for both community-dwelling respondents and those residing in nursing homes.

We examined bivariate associations between each predictor and all-cause dementia using the method proposed by Fine and Gray [25] to account for the semi-competing risk of death. The Fine and Gray approach treats the cumulative incidence function as a subdistribution function which can be defined at time $t$ as the instantaneous rate of occurrence of event $k$ among respondents who have not experienced an event of that type prior to time $t$ [25]. This allows one to model the effects of covariates on the cumulative incidence function in the presence of competing risks, producing subdistribution hazard ratios (sdHRs). This approach accounts for the fact that respondents who die prior to incident dementia are no longer considered at risk for dementia, as opposed to estimators such as Kaplan-Meier which treat the semi-competing risk of death as noninformative censoring which may bias results and overestimate associations between risk factors and dementia [13]. Respondents were observed from baseline until incident dementia, death, or censoring. We used age as the timescale (i.e., age at first dementia diagnosis, age at death, age at last visit) due to its strong association with dementia. All models were stratified by race/ethnicity and gender due to known differences in longevity and dementia risk across these strata [26, 27]. We separately examined bivariate associations between each predictor and dementia following the same procedures as described above but using cause-specific hazard models as recommended by Latouche and colleagues [28]. More details on the Fine and Gray model are provided in the S2 Appendix in S1 File.

We then used random survival forests for competing risks [29] to simultaneously investigate the relative importance of each predictor for dementia across racial/ethnic and gender groups while accounting for the semi-competing risk of death. Age was used as the timescale. Random survival forests are an extension of the random forest algorithm, an ensemble-based classification method which fits a series of classification and regression trees and then pools results across the trees [30]. We implemented this approach using the R package 'randomForestSRC' with 1,000 trees [31]. More details on the random survival forests procedure is provided in the S2 Appendix in S1 File.

## Results

The analytic sample for the primary analysis was comprised of 7,908 respondents with an average age at baseline of 65.6 years (standard error = 0.15). At baseline, 299 (3.7%) respondents were surveyed by proxy. Overall, 37.4% of respondents were non-Hispanic white men; 49.9% of respondents were non-Hispanic white women; 4.5% of respondents were non-Hispanic black men; and non-Hispanic black women comprised 8.2% of the sample. Summary characteristics for the analytic HRS sample at baseline are shown in S1 Table in S1 File. Correlations between all predictors by race/ethnicity and gender are presented in S1 Fig.

Fig 1 and S2 Table in S1 File present the sdHRs and 95% confidence intervals (CIs) for each risk factor on dementia examined independently in Fine and Gray models stratified by race/

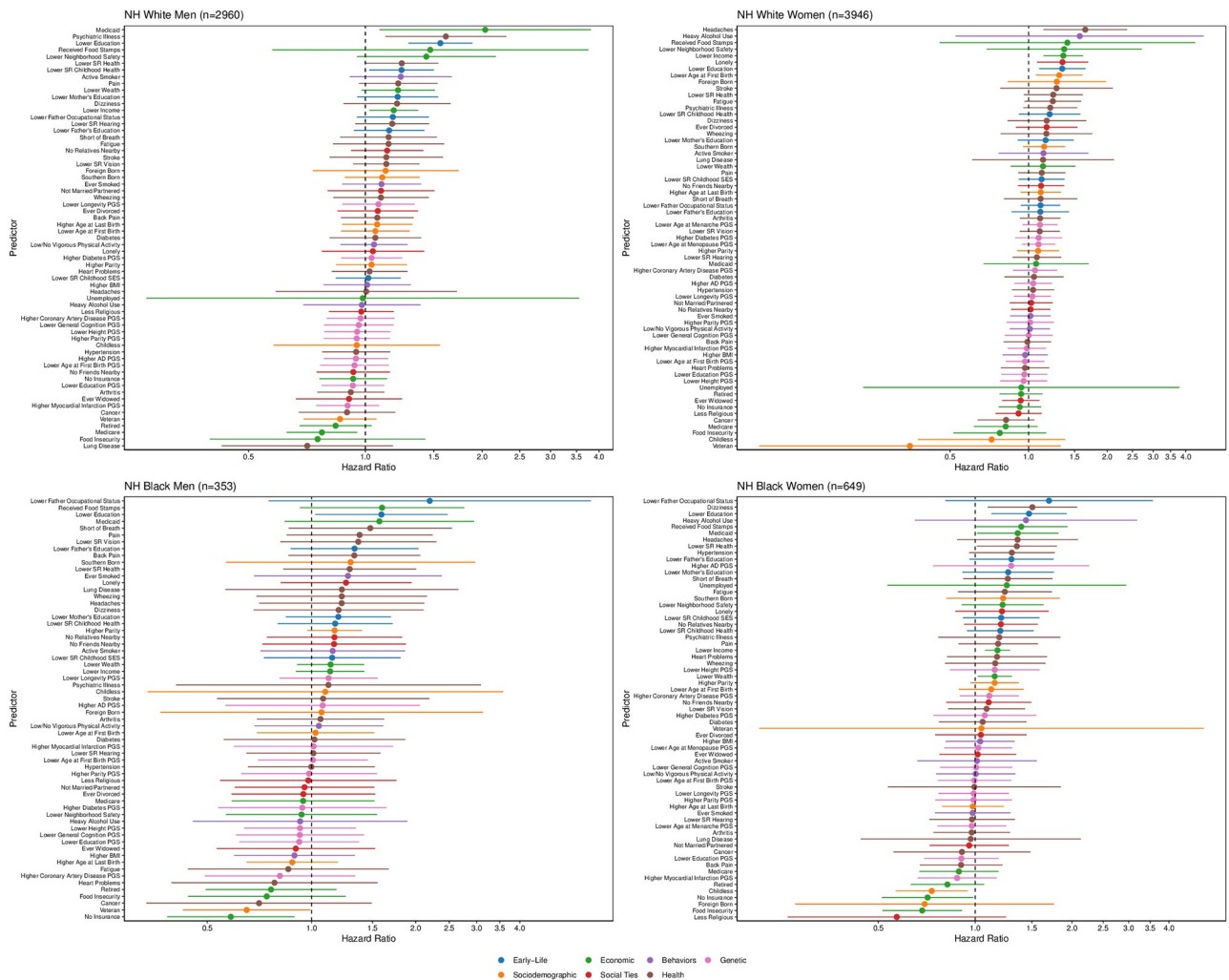

**Fig 1. Subdistribution Hazard Ratios (sdHRs) and 95% Confidence Intervals (CI) of each predictor for incident dementia obtained from Fine and Gray regression models stratified by race and gender.** Predictors with sdHRs equal to zero are excluded from the figure but retained in S3 Table in S1 File. Models use full analytic sample and classify dementia using the Langa-Weir classification scheme.

ethnicity and gender. Risk factors are categorized by domain and ranked from largest increase in risk of dementia (top of Figure) to largest decrease in risk of dementia (bottom of Figure). Risk factors with CIs that cross one are not considered statistically significant at the P value = 0.05 level.

Three of the top 10 characteristics—as determined by the magnitude of the sdHRs—for non-Hispanic white men and women were consistent: lower education, lower neighborhood safety, received food stamps. Among the top 10 characteristics for non-Hispanic white and black men, four overlapped: lower education, received food stamps, reported pain, reported Medicaid. Four of the top 10 characteristics for non-Hispanic white and black women were consistent: lower education, received food stamps, heavy alcohol use, and reported headaches. Across racial/ethnic and gender groups, lower education and receipt of food stamps were the only characteristics that consistently ranked in the top 10 of predictors. Receipt of food stamps was the only characteristic that consistently ranked in the top five of predictors of all racial/ethnic and gender groups. These results were consistent when comparing HRs obtained from

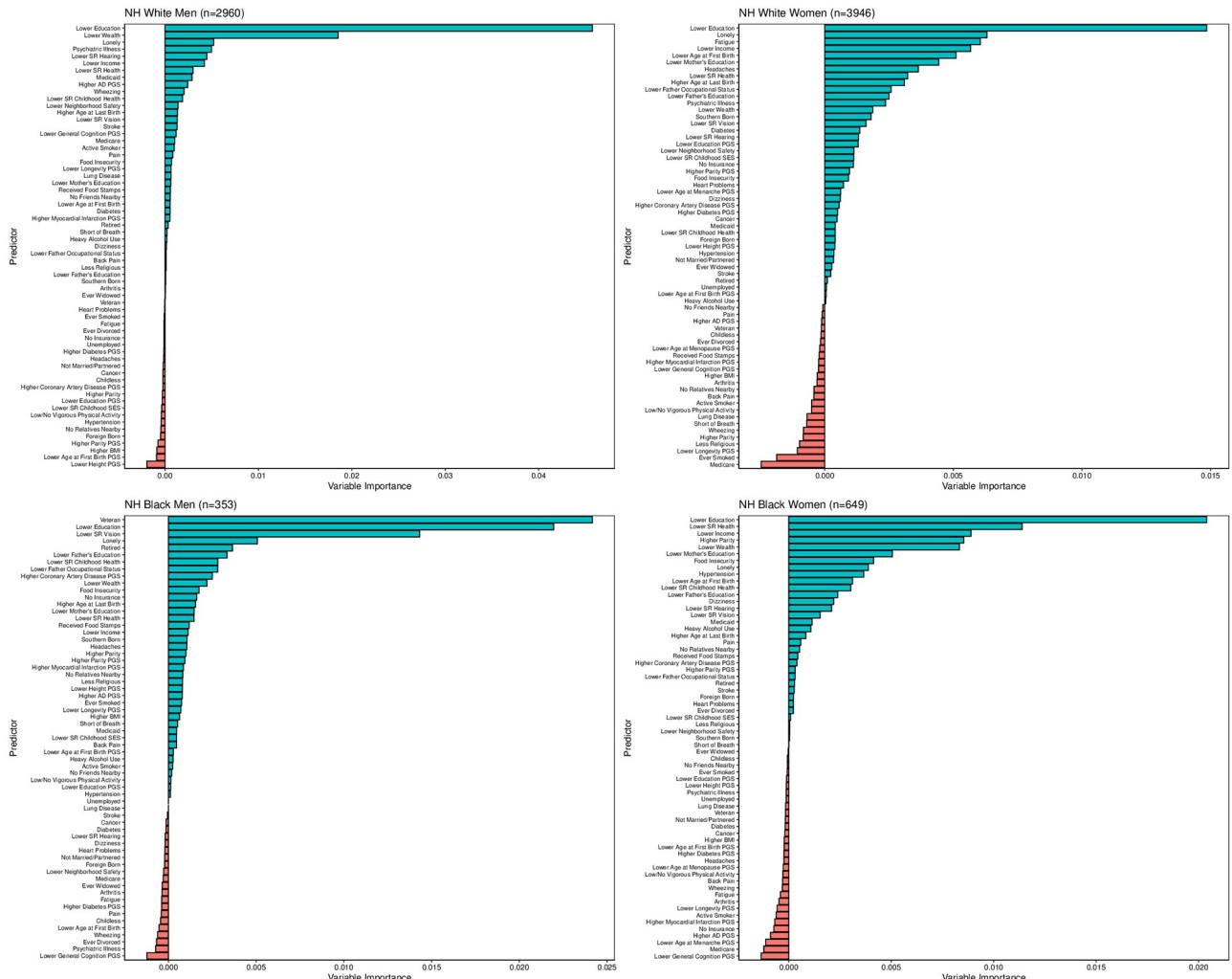

**Fig 2. Variable importance plot for 65 characteristics predicting dementia obtained from random survival forests for competing risks stratified by race and gender.** Model uses full analytic sample and classifies dementia using the Langa-Weir classification scheme.

cause-specific hazard models (S3 Table in S1 File) with the exception of lower education, which did not rank in the top 10 of predictors for non-Hispanic white women. However, it is evident that the point estimates and CIs for most predictors overlap across racial/ethnic and gender groups as shown in S2 Fig suggesting that the differences in the associated hazards across racial/ethnic groups are not statistically significant at the P value = 0.05 level.

Results from the random survival forests analysis for competing risks are shown in Fig 2. Blue bars indicate predictors with positive variable importance values; red bars indicate predictors with negative variable importance values, which in this context can be considered statistically insignificant. The positive length of the bar indicates the importance of each predictor. The top predictors across racial/ethnic and gender groups differed from those obtained in the Fine and Gray models, and were more consistent across racial/ethnic groups in the random survival forests analysis. Whereas lower education and lower neighborhood safety were the only consistently ranked predictors in the top 10 across race/ethnicity and gender groups in the Fine and Gray models, lower education and loneliness were consistently ranked in the top 10 in the random survival forests analysis. Lower age at first birth

and lower levels of respondent's mother's education appeared in the top 10 for non-Hispanic white and black women. Lower income and self-reported health ranked in the top 10 for all groups with the exception of non-Hispanic black men whereas lower wealth ranked in the top 10 for all groups with the exception of non-Hispanic white women. Fig 3 shows the rank order for predictors, overall, and for each race/ethnicity and gender group. The overall rank order was determined by calculating the unweighted mean of predictor rank orders within each of the four demographic strata, and sorting from lowest to highest mean (i.e., highest rank to lowest rank). The values of the overall rank order in Fig 3 do not represent the means themselves but instead correspond to these rankings. These results illustrate the variation across and within racial/ethnic groups. For example, whereas lower wealth is respectively ranked as the second and fifth leading predictor for non-Hispanic white men and non-Hispanic black women, lower wealth ranked 13th for non-Hispanic white women and 10th for non-Hispanic black men. Food insecurity, which ranked 10th overall, was ranked 20th and 23rd for non-Hispanic white men and women, but 11th and 7th for non-Hispanic black men and women.

In sensitivity analyses among a subsample of 6,746 respondents who were 70+ years of age at baseline and had at least one measure of all four classification schemes for dementia, we observed that the consistency of predictors with the highest sdHRs from the Fine and Gray models varied by race/ethnicity and gender group. For non-Hispanic white men, four of the top five predictors were consistent across all four classification schemes (lower education, lower neighborhood safety, receipt of food stamps, Medicaid, and psychiatric illness; S3–S6 Figs and S4–S7 Tables in S1 File). Among non-Hispanic white women, only headaches were among the top five predictors with the highest sdHRs across all four classification schemes (S3–S6 Figs). For non-Hispanic black men, only Medicaid was consistently ranked in the top five predictors with the highest sdHRs and among non-Hispanic black women, self-reported persistent dizziness was the only predictor consistently ranked in the top five (S3–S6 Figs). For the random survival forests analysis, three of the top five predictors (lower education, lower income, loneliness) were consistent overall when using the four different classification schemes (S7–S13 Figs). When used as the outcome in the Fine and Gray and random survival forests analyses, the Langa-Weir classifier, which does not account for race/ethnicity, gender, or education, resulted in more socioeconomic risk factors being ranked in the top five. Models which used the Hurd, Expert, and LASSO classifiers, which do account for respondent demographics, were more likely to produce health, behavioral, and genetic risk factors as top predictors.

## Discussion

In this 14-year population-based study of older adults in the US with available polygenic score data, we found that the relative importance of risk factors for predicting dementia varied across racial/ethnic and gender groups using two distinct methodologies. We also found the predictor rankings to vary based on the type of dementia classification used. Although not all observed differences were statistically significant, our stratified models may offer insight into substantive differences in the relative importance of risk factors across racial/ethnic and gender groups.

We observed variation in the rank order of predictors across and within racial/ethnic and gender groups in both the Fine and Gray models and random survival forests. The consistency of our primary results across both analyses suggests that our findings are robust to these two distinct approaches. However, in our sensitivity analyses in which we compared four

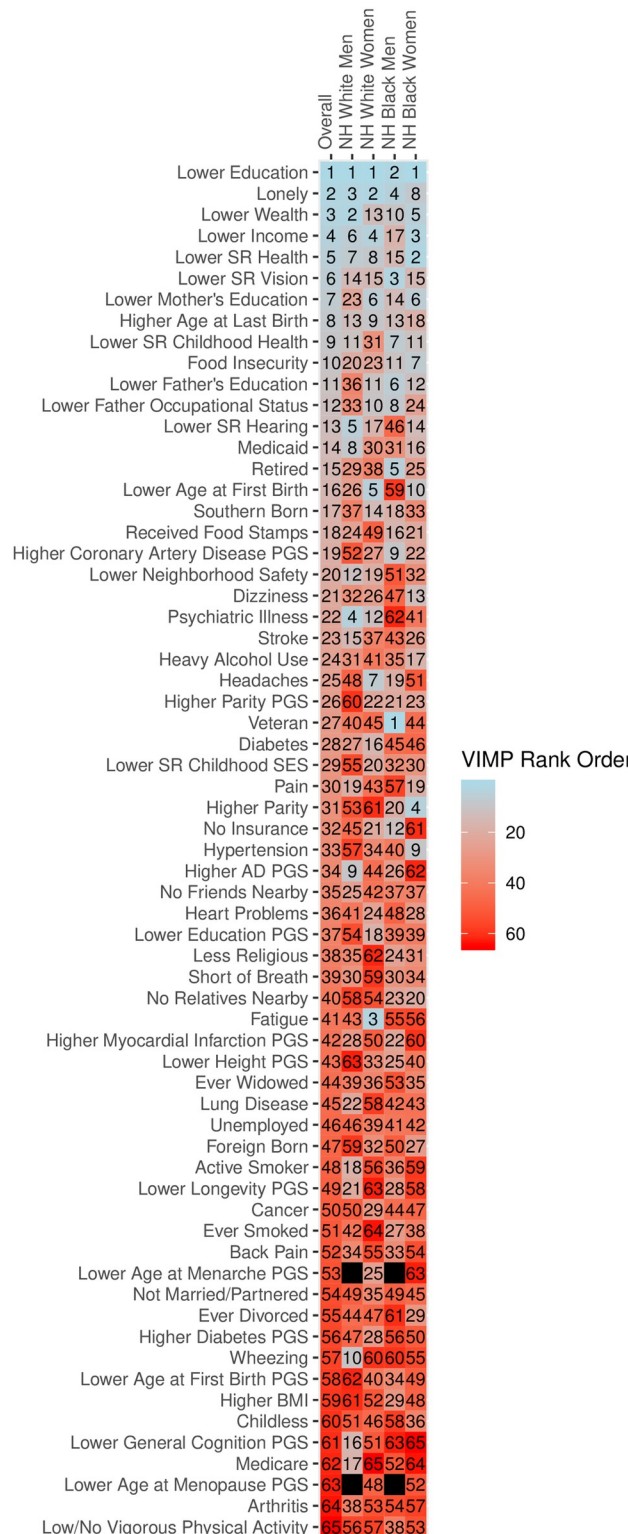

**Fig 3. Rank order of predictors obtained from random survival forests for competing risks stratified by race and gender.** Model uses full analytic sample and classifies dementia using the Langa-Weir classification scheme.

alternative classification schemes for dementia, we found the results to vary which may be due to the different criteria used in each of these classification schemes for dementia.

The low ranking of genetic predictors was apparent across our analyses, whereas we saw stronger associations between characteristics measured in mid- or later-life particularly those in the economic domain. Lower levels of education and receipt of food stamps were the only characteristics that consistently ranked in the top 10 of predictors for all groups in the random survival forests analysis. There was heterogeneity within and outside of the top 10 predictors highlighting the importance of identifying effective methods to promote health and mitigate dementia burden and its risk factors within racial/ethnic and gender groups. The results obtained in our analyses were similar to those reported by Casanova and colleagues [11] and Aschwanden et al. [3] but our account of the competing risk of mortality and stratification by race/ethnicity and gender may offer more accurate estimates and provide additional insight into the differential ranking of risk factors within these groups.

In their recent study, Aschwanden and colleagues [3] used Cox proportional hazard models and a machine-learning approach to predict cognitive impairment and dementia in the HRS over a 10-year period. The authors used 52 multi-domain risk factors in their random survival forests analysis and found that increases in body mass index, higher levels of emotional distress, diabetes, self-reporting race as black, and higher reports of childhood trauma were the top five predictors of dementia over the study period. The authors incorporated two polygenic scores—one for Alzheimer's disease which included the apolipoprotein e4 allele and one which did not. The authors then examined a subset of top predictors from their random survival forests model in a semi-parametric survival analysis framework using Cox proportional hazards model. Of the six predictors the authors examined, only emotional distress was significantly associated with incident dementia.

Despite similar methodologies and data, none of the top five predictors from Aschwanden and colleagues' study appeared in the overall top 10 from our random survival forests model and only two risk factors (education and self-reported childhood health) ranked in both top 10 lists. In our random survival forests analysis which accounted for the competing risk of death, we found that psychiatric illness—which is different from but most comparable to the author's emotional distress measure—ranked fourth among non-Hispanic white men, 12th among non-Hispanic white women, 62nd among non-Hispanic black men, and 41st among non-Hispanic black women. Interestingly, however, the sdHR for emotional distress obtained from Aschwanden and colleagues' study (sdHR: 1.85; 95%CI: 1.41, 2.44) overlapped with the sdHR for psychiatric illness among non-Hispanic white men in our independent Fine and Gray model (sdHR: 1.61; 95%CI: 1.13, 2.31) whereas the sdHR for psychiatric illness was not statistically significant for any other subgroup in our analysis. Moreover, the prior study reported increasing trajectories of body mass index as the top predictor of dementia in their random survival forests analysis whereas in our study, which used baseline body mass index in the year 2000, body mass index was ranked overall as the 59th predictor out of 65 and at best, ranked 29th for non-Hispanic black women. The relationship between body mass index and dementia is complex [32–34], with investigators reporting in one study that midlife obesity was associated with an increased risk of dementia compared to those with normal body mass index whereas this association reversed in later-life [33]. It is possible that, by not accounting for the competing risk of mortality, the top predictors reported in Aschwanden and colleagues' study are driven by their association with mortality although we did not test this directly.

There are several hypotheses linking educational attainment and experiences over the life course to cognitive impairment in later adulthood. Although studies exploring the potential mechanisms linking these risk factors to dementia are inconclusive, an expanding body of work has investigated the cognitive reserve hypothesis which suggests that there are individual

differences in the ability to cope with brain pathology [35, 36]. Educational attainment, along with socioenvironmental exposures at different stages in the life course and genetic makeup may play an important role in fostering brain development which may translate to a healthier, more resilient brain in older adulthood. In a recent study by Xu and colleagues [37], for example, the authors found that higher lifespan cognitive reserve—measured by educational attainment, life course cognitive activities, and social activities in older adulthood—was associated with a reduced risk of dementia. Further, the authors noted a dose-dependent association between cognitive reserve and dementia risk which was evident even in the presence brain pathology (e.g., β-amyloid plaques).

As noted above, a major strength of this study is our account of the competing risk of death. Recent studies have reported that studying age-related conditions, including dementia, while not accounting for the competing risk of death may produce biased or misleading results [38, 39]. An additional strength of our study is the inclusion of 65 risk factors spanning socio-demographic, early-life, economic, health and behavioral, social, and genetic risk factor domains across the life course. Our inclusion of genetic risk factors was in the form of poly-genic scores which are able to capture genotypic variation across multiple genetic loci compared to individual genotypes. In addition, we compared our primary results using the Langa-Weir classification scheme to more recent approaches which may be better suited to studying disparities in dementia across racial/ethnic groups as well as among adults who vary with respect to socioeconomic status [19, 20].

This study also had several limitations. First, there were several risk factors in the report by the Lancet Commission on Dementia Prevention and Care that are not available in the HRS. These measures include for example, dietary quality, exposure to environmental contaminants, and questions about cognitive training and stimulation. Second, although we used clinically validated cut points derived from the ADAMS for assessing dementia in the HRS cohort, these classifications may be subject to measurement error. This measurement error could affect the coefficient estimates and rankings if a large enough sample of respondents were misclassified with respect to their cognitive status. We conducted sensitivity analyses using three alternative classification schemes for dementia which may alleviate some of this concern. Third, as with any tree-based approach such as the random forest algorithm, variables with a wider range and therefore more points at which they can be split, tend to have higher predictive power [40]. We addressed this limitation by standardizing continuous measures to make them as equivalent as possible across our study.

We identified heterogeneity in the association between dementia and its risk factors across racial/ethnic and gender groups using a more traditional approach to account for competing risks (the Fine and Gray model) as well as a more contemporary data-driven approach (random survival forests for competing risks). These results may be useful for understanding and further exploring recent reports documenting disparity trends in dementia across racial/ethnic and gender groups [7, 41, 42]. We advise caution in treating these results with a causal interpretation and instead suggest that these results can be used for hypothesis generation and to inform future observational and clinical studies to identify the multiple pathways through which these risk factors may be differentially associated with the risk of dementia across demographic strata.

## Supporting information

**S1 Fig. Correlation matrices for 65 predictors stratified by race and gender.**
(PDF)

**S2 Fig. Comparison of Subdistribution Hazard Ratios (sdHRs) and 95% Confidence Intervals (CI) of each predictor for incident dementia obtained from Fine and Gray regression models stratified by race and gender.** Model uses full analytic sample and classifies dementia using the Langa-Weir classification scheme.
(PDF)

**S3 Fig. Cause-specific Hazard Ratios (HRs) and 95% Confidence Intervals (CI) of each predictor for incident dementia obtained from cause-specific hazards regression models stratified by race and gender.** Models use restricted analytic sample and classify dementia using the Langa-Weir classification scheme. Predictors with HRs equal to zero are excluded from the figure but retained in S5 Table in S1 File.
(PDF)

**S4 Fig. Subdistribution Hazard Ratios (sdHRs) and 95% Confidence Intervals (CI) of each predictor for incident dementia obtained from Fine and Gray regression models stratified by race and gender.** Models use restricted analytic sample and classify dementia using the Hurd classification scheme. Predictors with HRs equal to zero are excluded from the figure but retained in S6 Table in S1 File.
(PDF)

**S5 Fig. Subdistribution Hazard Ratios (sdHRs) and 95% Confidence Intervals (CI) of each predictor for incident dementia obtained from Fine and Gray regression models stratified by race and gender.** Models use restricted analytic sample and classify dementia using the Expert classification scheme. Predictors with HRs equal to zero are excluded from the figure but retained in S7 Table in S1 File.
(PDF)

**S6 Fig. Subdistribution Hazard Ratios (sdHRs) and 95% Confidence Intervals (CI) of each predictor for incident dementia obtained from Fine and Gray regression models stratified by race and gender.** Models use restricted analytic sample and classify dementia using the LASSO classification scheme. Predictors with HRs equal to zero are excluded from the figure but retained in S8 Table in S1 File.
(PDF)

**S7 Fig. Variable importance plot for 65 characteristics predicting dementia obtained from random survival forests for competing risks stratified by race and gender.** Model uses restricted analytic sample and classifies dementia using the Langa-Weir classification scheme.
(PDF)

**S8 Fig. Variable importance plot for 65 characteristics predicting dementia obtained from random survival forests for competing risks stratified by race and gender.** Model uses restricted analytic sample and classifies dementia using the Hurd classification scheme.
(PDF)

**S9 Fig. Variable importance plot for 65 characteristics predicting dementia obtained from random survival forests for competing risks stratified by race and gender.** Model uses restricted analytic sample and classifies dementia using the Expert classification scheme.
(PDF)

**S10 Fig. Variable importance plot for 65 characteristics predicting dementia obtained from random survival forests for competing risks stratified by race and gender.** Model uses restricted analytic sample and classifies dementia using the LASSO classification scheme.
(PDF)

**S11 Fig. Rank order of predictors obtained from random survival forests for competing risks stratified by race and gender.** Model uses restricted analytic sample and classifies dementia using the Langa-Weir classification scheme.
(PDF)

**S12 Fig. Rank order of predictors obtained from random survival forests for competing risks stratified by race and gender.** Model uses restricted analytic sample and classifies dementia using the Hurd classification scheme.
(PDF)

**S13 Fig. Rank order of predictors obtained from random survival forests for competing risks stratified by race and gender.** Model uses restricted analytic sample and classifies dementia using the Expert classification scheme.
(PDF)

**S14 Fig. Rank order of predictors obtained from random survival forests for competing risks stratified by race and gender.** Model uses restricted analytic sample and classifies dementia using the LASSO classification scheme.
(PDF)

**S1 File.**
(DOCX)

## Author Contributions

**Conceptualization:** Jordan Weiss, Eli Puterman, Aric A. Prather, Erin B. Ware, David H. Rehkopf.

**Data curation:** Jordan Weiss, David H. Rehkopf.

**Formal analysis:** Jordan Weiss, David H. Rehkopf.

**Investigation:** Jordan Weiss, David H. Rehkopf.

**Methodology:** Jordan Weiss, Eli Puterman, Aric A. Prather, Erin B. Ware, David H. Rehkopf.

**Software:** Jordan Weiss, David H. Rehkopf.

**Writing – original draft:** Jordan Weiss, Eli Puterman, Aric A. Prather, David H. Rehkopf.

**Writing – review & editing:** Jordan Weiss, Eli Puterman, Aric A. Prather, Erin B. Ware, David H. Rehkopf.

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
