## [Decision Letter · Decision Letter 0]

9 Jul 2020

PONE-D-20-16868

A data-driven prospective study of dementia among older adults in the United States

PLOS ONE

Dear Dr. Weiss,

Thank you for submitting your manuscript to PLOS ONE. After careful consideration by 2 Reviewers and an Academic Editor, all of the critiques of both Reviewers must be addressed in detail in a revision to determine publication status. If you are prepared to undertake the work required, I would be pleased to reconsider my decision, but revision of the original submission without directly addressing the critiques of the two Reviewers does not guarantee acceptance for publication in PLOS ONE. If the authors do not feel that the queries can be addressed, please consider submitting to another publication medium. A revised submission will be sent out for re-review. The authors are urged to have the manuscript given a hard copyedit for syntax and grammar.

**Comments to the Author**

1. Is the manuscript technically sound, and do the data support the conclusions?

Reviewer #1: Yes

Reviewer #2: Yes

2. Has the statistical analysis been performed appropriately and rigorously? 

Reviewer #1: Yes

Reviewer #2: Yes

3. Have the authors made all data underlying the findings in their manuscript fully available?

Reviewer #1: Yes

Reviewer #2: Yes

4. Is the manuscript presented in an intelligible fashion and written in standard English?

Reviewer #1: Yes

Reviewer #2: Yes

5. Review Comments to the Author

Reviewer #1: This manuscript explores what risk factors across broad and diverse research fields may be most important to predicting dementia. This is a rigorous examination of a range of measures in the HRS data that may predict dementia, for different race and sex groups, using robust statistical analyses that account for the competing risk of death. The top five predictors across all groups were lower education, loneliness, lower wealth and income, and lower self-reported health, and the authors did find variation in the leading predictors of dementia across racial/ethnic and gender groups. The different measures used to determine a classification of dementia in these data did change the order of predictors.

There are so many predictors in each of the figures for each of the sex/race groups, the text has to be so small to include them all in one figure, it makes these figures difficulty to read. The authors might want to consider placing the sensitivity analysis / supplemental figures in the actual supplement and not in the primary manuscript.

Other minor issues include:

1. No page numbers visible in the manuscript.

2. Page 15, line 178 looks like an incomplete sentence.

3. I’m not clear on why the dichotomous variables are coded -1/1 instead of 0/1. Is this a standard in machine learning? Or in the Fine and Gray method?

4. How does the Fine and Gray “semi-competing risk of death” differ from the "competing risk of death" discussed in other projects following the Fine and Gray method?

Reviewer #2: 

1. Methodological strengths include large sample size, age range (50-), stratification by race/ethnicity and gender, machine learning prediction models (RSF), 65 and risk factors from multiple (8) life history facets.

2. The importance of data-driven analytics to this area (as complementary to hypothesis-guided, single risk factor analyses and reviews) is carefully established in the introduction. One article the authors seem to be missing had a much smaller sample size but included a much broader range of predictors and, along with normal and AD groups, an MCI group (Sapkota et al., 2018, doi.10.3389/fnagi.2018.00296).

3. In addition to methodological strengths, some limitations accompany the HRS data set for this study. The limitations begin with the outcome measures and diagnostic procedures. On line 125, the outcome is comprised of three simple cognitive measures, not very useful as cognitive outcomes per se, but also not typically used for diagnostic purposes. Moreover, some of the participants (indicate n in this section) performed these tasks whereas others were evaluated informally by “respondents”. These informal reports were not matched to the three actual performance measures. A “cut point” (including validation and sensitivity) procedure is described (line 138-150), but both the input (the cognitive performance and the subjective cognitive evaluations) and the eventual output (dementia classification) are under-described, unclear in their replicability or validity. No note is made of the type of dementia classified.

4. The risk factor data were collected in 1998 or 2000. When were the cognitive data collected? Follow-ups to 2014 are mentioned, but longitudinal analyses are not mentioned. Clarify design in the ms.

5. Line 178 seems to be missing something in the last sentence.

6. The risk factor prediction differences across race/ethnicity and gender are interesting. The reporting style (beginning on line 221) is difficult to follow and has little integration and no theoretical/mechanistic interpretation. What criteria were used for selecting the top 10 predictors (and were they the same across the groups)? The listing in this section (221-258) could be more informatively presented.

7. Readers and reviewers will await an integration in the discussion, but the interpretation in this section is at a level of comparison across stratification groups and with some other studies (especially reference 3, but not linked back to more general review of risk factors, such as that in reference 1). I did not see an attempt to identify potential mechanisms associated with any of the identified predictors. What differential processes do these risk factors represent? One that stood out was “food stamps” but all deserve some attention in terms of what they mean.

8. In the discussion (line 322f) the rank orders (including as low as 62nd) are noted as though they have some comparative significance for interpretation. Is it valid to use these ordinal rankings interpretively, without qualification?

9. The genetic risk factors did not perform well. Did you try APOE alone?

6. PLOS authors have the option to publish the peer review history of their article (what does this mean?). If published, this will include your full peer review and any attached files.

**Do you want your identity to be public for this peer review?** For information about this choice, including consent withdrawal, please see our Privacy Policy.

Reviewer #1: No

Reviewer #2: No

We look forward to receiving your revised manuscript.

Kind regards,

Stephen D. Ginsberg, Ph.D.

Section Editor

PLOS ONE

---

## [Author Response · Author response to Decision Letter 0]

13 Aug 2020

“A data-driven prospective study of dementia among older adults in the United States”

(First Revision of PLOS One Manuscript ID: [PONE-D-20-16868])

POINT-BY-POINT RESPONSE

We thank the editor and the two anonymous referees for their very helpful comments and recommendations on our manuscript. We have revised our manuscript to address their comments and to comply with the journal formatting standards. Below, we provide a detailed response to each comment as well as a description of the corresponding revisions. Reviewer comments are italicized whereas our responses are in normal font.

Reviewer 1

R1.0. There are so many predictors in each of the figures for each of the sex/race groups, the text has to be so small to include them all in one figure, it makes these figures difficulty to read. The authors might want to consider placing the sensitivity analysis / supplemental figures in the actual supplement and not in the primary manuscript.

R1.0 Response. We thank the reviewer for their suggestion. We revised the organization of our original submission and now feature the sensitivity analysis and supplemental figures in the actual supplement. 

R1.1. No page numbers visible in the manuscript.

R1.1 Response. The revised manuscript now features page numbers centered at the bottom of every page.

R1.2. Page 15, line 178 looks like an incomplete sentence.

R1.2 Response. We have removed the incomplete sentence that appeared on page 15, line 178 of the original submission.

R1.3. I’m not clear on why the dichotomous variables are coded -1/1 instead of 0/1. Is this a standard in machine learning? Or in the Fine and Gray method?

R1.3 Response. We thank the reviewer for their inquiry. We note that the continuous variables were standardized with mean 0 and standard deviation 1. Coding a binary variable in the typical 0/1 could inflate the effect size relative to a standardized variable with mean 0 and standard deviation 1. For example, assuming a variable has an equal distribution of 0s and 1s, the mean and standard deviation would be 0.5 if the sample were sufficiently large. To remedy this, we coded the binary variables as -1/1 which, following from the example, would result in mean 0 and standard deviation 1 in a sufficiently large sample, mitigating concerns about comparability.

R1.4. How does the Fine and Gray “semi-competing risk of death” differ from the "competing risk of death" discussed in other projects following the Fine and Gray method?

R1.4 Response. We thank the reviewer for their inquiry. A semi-competing risks framework is one in which the focus of study is on a non-terminal event (in our case, dementia) whose occurrence may be subject to a terminal event (i.e., death). Because incident dementia does not inhibit death, dementia itself is not a competing event for death. Thus, since incident dementia does not “compete” with death, we have a semi-competing risks framework. If our study were focused on dementia-specific mortality versus all other causes of death, we would have a [fully] competing risks framework because one outcome would inhibit the other.

Reviewer 2

R2.1. Methodological strengths include large sample size, age range (50-), stratification by race/ethnicity and gender, machine learning prediction models (RSF), 65 and risk factors from multiple (8) life history facets.

R2.1 Response. We thank the reviewer for their favorable views on the manuscript.

R2.2. The importance of data-driven analytics to this area (as complementary to hypothesis-guided, single risk factor analyses and reviews) is carefully established in the introduction. One article the authors seem to be missing had a much smaller sample size but included a much broader range of predictors and, along with normal and AD groups, an MCI group (Sapkota et al., 2018, doi.10.3389/fnagi.2018.00296).

R2.2 Response. We thank the reviewer for sharing this reference. We excluded this article from the original submission because it is focused on Alzheimer’s disease which represents a fractional majority of dementia cases. However, we decided to reference the suggested article to provide a more thorough review of the literature in our revised manuscript.

R2.3. In addition to methodological strengths, some limitations accompany the HRS data set for this study. The limitations begin with the outcome measures and diagnostic procedures. On line 125, the outcome is comprised of three simple cognitive measures, not very useful as cognitive outcomes per se, but also not typically used for diagnostic purposes. Moreover, some of the participants (indicate n in this section) performed these tasks whereas others were evaluated informally by “respondents”. These informal reports were not matched to the three actual performance measures. A “cut point” (including validation and sensitivity) procedure is described (line 138-150), but both the input (the cognitive performance and the subjective cognitive evaluations) and the eventual output (dementia classification) are under-described, unclear in their replicability or validity. No note is made of the type of dementia classified.

R2.3 Response. We thank the reviewer for their comment. We agree that the ascertainment of dementia in the Health and Retirement Study has its limitations as there is limited clinical information for all respondents. However, as a nationally representative, longitudinal, population-based prospective study, the Health and Retirement Study also has several strengths. One of these strengths is a supplement to the Health and Retirement Study known as the Aging, Demographics and Memory Study (ADAMS) which involved nurses and psychometric technicians travelling to a stratified random subsample of Health and Retirement Study respondent’s homes to conduct in-depth cognitive assessments that took approximately 3-4 hours. A final diagnosis of dementia (including possible and probable Alzheimer’s Disease [AD], vascular dementia, etc.) was made by a consensus panel comprised of an expert team of neurologists, psychiatrists, neuropsychologists, and internists. The 3-4 hour cognitive battery used among the ADAMS sample included the cognitive battery administered to the complete Health and Retirement Study. Health and Retirement Study investigators Langa and Weir (Langa, Kabeto, & Weir, 2009) used these samples to develop cut-points for the Health and Retirement Study cognitive measures that would produce the same population distribution of cognitive states estimated by ADAMS (i.e., “equipercentile equating”).

Among self-respondents, cognitive status was assessed using immediate and delayed recall and a serial 7s subtraction task, in addition to a backwards counting task. The first three of these tasks are also present in the Mini-Mental State Exam which, despite its imperfections, is among the most widely used screeners for dementia. 

Cognitive status among proxy respondents (a spouse or other family member of the respondent) was assessed using 16 questions about the respondent’s change in memory for various types of information with regard to change in the last two years. These questions are adapted from the short form of the Informant Questionnaire on Cognitive Decline in the Elderly (IQCODE; Jorm, 1994).

In the revised manuscript, we (i) specified the number and sample-weighted percentage of respondents surveyed by proxy at baseline, (ii) clarified that our outcome is all-cause dementia, and (iii) provided a reference for readers who are interested in the details surrounding the cognitive measures in the Health and Retirement Study.

R2.4. The risk factor data were collected in 1998 or 2000. When were the cognitive data collected? Follow-ups to 2014 are mentioned, but longitudinal analyses are not mentioned. Clarify design in the ms.

R2.4 Response. We thank the reviewer for raising this point which we have clarified in the revised manuscript with the following statement: “Dementia status for self- and proxy-respondents was assessed at each survey wave.” The Health and Retirement Study is a longitudinal biennial survey and the cognitive assessments occur with every survey wave for all respondents.

R2.5. Line 178 seems to be missing something in the last sentence.

R2.5 Response. We have removed the incomplete sentence that appeared on line 178 of the original submission.

R2.6. The risk factor prediction differences across race/ethnicity and gender are interesting. The reporting style (beginning on line 221) is difficult to follow and has little integration and no theoretical/mechanistic interpretation. What criteria were used for selecting the top 10 predictors (and were they the same across the groups)? The listing in this section (221-258) could be more informatively presented.

R2.6 Response. We thank the reviewer for highlighting the issues with our reporting style beginning on line 221. We have modified this section of the manuscript to remedy this issue. We also specified the criteria used to select the top 10 predictors with the following statement: “Three of the top 10 characteristics—as determined by the magnitude of the sdHRs—for non-Hispanic white men and women were consistent: lower education, lower neighborhood safety, received food stamps.”

R2.7. Readers and reviewers will await an integration in the discussion, but the interpretation in this section is at a level of comparison across stratification groups and with some other studies (especially reference 3, but not linked back to more general review of risk factors, such as that in reference 1). I did not see an attempt to identify potential mechanisms associated with any of the identified predictors. What differential processes do these risk factors represent? One that stood out was “food stamps” but all deserve some attention in terms of what they mean.

R2.7 Response. We thank the reviewer for their comment. Although elucidating pathways was not the primary objective of our study, we agree that identifying potential mechanisms would better inform the reader and could help guide future work. Thus, we provide in the discussion a brief overview of some of the top risk actors and how they may relate to dementia.

R2.8. In the discussion (line 322f) the rank orders (including as low as 62nd) are noted as though they have some comparative significance for interpretation. Is it valid to use these ordinal rankings interpretively, without qualification?

R2.8 Response. We thank the reviewer for their inquiry. The presented rankings are for illustrative purposes only. They can be compared and interpreted qualitatively.

R2.9. The genetic risk factors did not perform well. Did you try APOE alone?

R2.9 Response. We thank the reviewer for their inquiry. Due to the nature in which the Health and Retirement Study investigators prepared and released the genetic data, the polygenic score data are available from 2006-2012 whereas individual genetic data are—at this time—only available from 2006-2010. We discussed incorporating APOE alone as a risk factor but doing so would result in a reduction of our sample size by approximately 25%. We agree that there is value and merit to looking at APOE independently but decided against it to preserve sample size and power for our stratified analyses.

---

## [Decision Letter · Decision Letter 1]

17 Sep 2020

A data-driven prospective study of dementia among older adults in the United States

PONE-D-20-16868R1

Dear Dr. Weiss,

We’re pleased to inform you that your manuscript has been judged scientifically suitable for publication and will be formally accepted for publication once it meets all outstanding technical requirements.

Kind regards,

Stephen D. Ginsberg, Ph.D.

Section Editor

PLOS ONE

Additional Editor Comments: Please make the punctuation changes suggested by Reviewer #1 and conduct a copyedit for syntax and grammar.

**Comments to the Author**

1. If the authors have adequately addressed your comments raised in a previous round of review and you feel that this manuscript is now acceptable for publication, you may indicate that here to bypass the “Comments to the Author” section, enter your conflict of interest statement in the “Confidential to Editor” section, and submit your "Accept" recommendation.

Reviewer #1: All comments have been addressed

2. Is the manuscript technically sound, and do the data support the conclusions?

Reviewer #1: Yes

3. Has the statistical analysis been performed appropriately and rigorously? 

Reviewer #1: Yes

4. Have the authors made all data underlying the findings in their manuscript fully available?

Reviewer #1: No

5. Is the manuscript presented in an intelligible fashion and written in standard English?

Reviewer #1: Yes

6. Review Comments to the Author

Reviewer #1: We appreciate the authors' explanations in response to our previous comments. Please be advised that linea 210 and 218 on page 19 ("more details on...") are missing a period.

7. PLOS authors have the option to publish the peer review history of their article (what does this mean?). If published, this will include your full peer review and any attached files.

Reviewer #1: No

---

## [Editor Report · Acceptance letter]

21 Sep 2020

PONE-D-20-16868R1 

A data-driven prospective study of dementia among older adults in the United States 

Dear Dr. Weiss:

I'm pleased to inform you that your manuscript has been deemed suitable for publication in PLOS ONE. Congratulations! Your manuscript is now with our production department. 

Kind regards, 

on behalf of

Dr. Stephen D. Ginsberg 

Section Editor

PLOS ONE